# Cross-culturally adapted psychological interventions for the treatment of depression and/or anxiety among young people: A scoping review

**Masuma Pervin Mishu** [1]*, **Lucy Tindall**[2], **Philip Kerrigan**[2], **Lina Gega**[2,3]

**1** Institute of Epidemiology and Health Care, University College London, London, United Kingdom, **2** Department of Health Sciences, Faculty of Sciences, University of York, York, United Kingdom, **3** Hull York Medical School, University of York, York, United Kingdom

\* masuma.mishu@ucl.ac.uk

## Abstract

### Background

Mental health problems among young people are a major global public health challenge. Psychological interventions may improve mental health, yet most are developed in western cultures, and it is unclear whether they are applicable to other geographical settings and can be delivered successfully to diverse populations. We identified empirical studies focusing upon cross-culturally adapted psychological interventions and examined the cultural adaptation process used and the effectiveness of the interventions in the treatment of depression and/or anxiety disorders among young people (defined here as children and adolescents aged between 8–18 years).

### Method

We conducted a scoping review aligning to the guidelines reported in the Preferred Reporting Items for Systematic Reviews and Meta-Analysis Extension for Scoping Reviews (PRISMA-ScR) Statement. Stakeholder engagement enabled us to discuss the findings of the review and obtain feedback.

### Results

We identified 17 studies of cross-culturally adapted psychological interventions that considered the appropriate language, metaphors, culturally appropriate terms, and cultural values of young people. Most studies (n = 11) adopted a randomised control trial (RCT) methodology. Six studies used the ecological validity and cultural sensitivity framework. Planned adaptation, cultural adaptation of content, and surface and deep structure level adaptations were used in other studies. Apart from one pilot study, all studies reported that culturally adapted interventions resulted in improvements in depression and/or anxiety symptoms in young people. The results suggest the potential effectiveness of cross-culturally adapted

**Data Availability Statement:** All relevant data are within the manuscript and its Supporting Information files.

**Funding:** The authors received no specific funding for this work.

**Competing interests:** The authors have declared that no competing interests exist.

interventions within this context. Our stakeholder consultations demonstrated that engaging different community-level stakeholders in the adaptation process was highly recommended.

## Conclusions

Whilst most included studies indicated improvements in depression and/or anxiety symptoms in young people following a cross-culturally adapted intervention, more work is needed in this area. In particular, focus should be placed upon identifying the dimensions of interventions that should be culturally adapted to make them acceptable, engaging and effective.

## Background

Mental health problems among young people are major public health concerns globally. According to the World Mental Health Report published by the World Health Organization (WHO) in 2022, approximately 8% of children, aged 5–9 years, and 14% of adolescents, aged 10–19, are affected by mental health disorders globally [1]. Youth is a phase of life when mental health problems can arise and contribute to morbidity and mortality [2, 3] causing a major global public health challenge [4]. This results in significant costs, including lost economic productivity, and an increased burden on health, education, social protection, and justice systems [5, 6]. Mental health problems reduce healthy social participation and impede young people's development. Furthermore, mental health problems in youth can increase the likelihood of psychiatric illness and disability in adulthood with around 50% of adult mental health problems originating in childhood and adolescence [3].

Two major common mental health problems in young people are anxiety and depression [7, 8]. According to the United Nations Children's Fund's (UNICEF) State of the World's Children 2021 report, the most common mental health disorders are anxiety and depression, accounting for over 40% of cases among adolescents [9]. Although fear and anxiety are regarded as part of normal, as well as necessary development, severe anxiety symptoms impair various daily functions in young people [7] and can be predictive of later mental health problems including mood disorders, and substance abuse [8, 10]. Major depression is a common, persistent, and debilitating condition that has been associated with elevated psychiatric comorbidity and increased risk for academic failure, interpersonal problems, suicide attempts, and legal problems in adolescents [11]. Moreover, research suggests that the early onset of depressive episodes in childhood and adolescence increases the likelihood of future depressive episodes [12]. It is estimated that 1.1% of 10–14-year-olds and 2.8% of 15-19-year-olds experience depression, whereas anxiety affects 3.6% and 4.6% of the same age groups respectively [13]. In the WHO European region, depression and anxiety disorders fall into the top five causes of overall disease burden among young people (as measured by disability-adjusted life years).

Evidence-based psychological interventions have been established for young people with anxiety and/or depression and are recommended according to National Institute for Health and Care Excellence (NICE) guidelines. Several studies, including systematic reviews have shown that cognitive behavioural therapy (CBT) is a promising and effective psychological treatment of anxiety disorders in children and adolescents [14, 15] as well as depression [16, 17]. CBT is recommended for the treatment of young people with depression and/or anxiety as well as Interpersonal Therapy (IPT), family therapy and psychodynamic psychotherapy

[18]. More recently several systematic reviews [19, 20] have explored the delivery of Behavioural Activation (BA), an alternative to CBT within this context.

Psychological interventions have the potential to improve the mental health of young people. Access to mental health services for young people in non-Western countries and amongst different ethnic minority group and migrated populations in Western countries lags significantly behind that of the native population in Western countries [4, 21]. Much of this can be explained by differences in the provision of infrastructure and funding and also in differing cultural attitudes towards mental health which may hamper the development of national and local mental health policy and discourage individuals and families from engaging with services. As most of the psychological interventions are developed in western settings, their content and mode of delivery is shaped by that cultural environment. It is questionable whether these interventions are applicable to people from different cultural environments, whether those in other (non-Western) countries [22] or particular minority populations in a (Western) country. Translating the materials into another language is unlikely to be enough to make them accessible and engaging for populations with very different cultural points of reference. Also, the existing mode of delivery may not be responsive to the needs and cultural practices of community.

In the intervention field therefore, adaptation involves the systematic modification of key characteristics, elements, and methods of delivery while still maintaining the core elements and theory of the intervention [23]. This admittedly presupposes that the core elements and principles of an intervention are universally applicable across all cultures, which may not be absolutely the case. It is however beyond the scope of this review to explore such a question. This systematic modification should consider language, culture, and context in such a way that evidence-based treatments or intervention protocols are compatible with the client's cultural patterns, meanings, and values [24]. Different theories and frameworks for the process of cross cultural adaptation have been developed and used. One of the more commonly used frameworks is the ecological validity and cultural sensitivity framework developed by Bernal and Sáez-Santiago [25], where adaptation posits eight dimensions (language, persons, metaphors, content, concepts, goals, methods, and context) to increase the ecological and external validity of a treatment. Some other cultural adaptation processes include but not limited to surface and deep structure levels adaptation [26, 27], the planned adaptation approach [28], and content cultural adaptation [29, 30].

A meta-analysis examining cultural adaptations of psychological interventions for a range of psychological disorders in children, adolescents, and adults, supported the efficacy of culturally adapted interventions in comparison to no, or other, interventions [21]. Although, evidence suggests the importance of providing culturally adapted interventions [8, 31, 32], psychological interventions for young people with ethnic and cultural sensitivity are scarce and dissemination of such interventions to diverse cultures is still at an experimental stage in the field of evidence-based psychological intervention.

## Aim

Through a scoping review we sought to identify and explore empirical research that involved psychological interventions adapted for use cross-culturally. As the research in this area has not been explored in depth before, the aim of this review was to identify available literature, to collect related information from available studies using a variety of methods and study designs, and to seek an answer to a broad and complex question rather than assessing or identifying a specific outcome of specific interventions or assessing the quality of the studies. We therefore conducted this scoping review rather than a systematic review.

Objectives:

1. Identify available studies that tested the cross-culturally adapted interventions for the treatment of depression and/or anxiety among young people.

2. To explore the cross-cultural adaptation process and frameworks used for the cultural adaptation.

3. To examine the effectiveness of these adapted interventions in the treatment of depression and/or anxiety disorders among young people.

In this study we defined young people as children / pre-adolescents and adolescents, aged between 8–18 years. We chose 18 as a cut-off as we wished to limit our study to young people of school age.

## Methods

A scoping review [33] was undertaken aligning to the guidelines reported in the Preferred Reporting Items for Systematic Reviews and Meta-Analysis Extension for Scoping Reviews (PRISMA-ScR) Statement [34].

### Information sources and search strategy

The following electronic databases were searched in February 2023: Ovid MEDLINE, PsycINFO, EMBASE, Cochrane database of systematic reviews and CINAHL using a strategy developed with the help of an information specialist and modified for each database (the search strategy can be found in S1 File and PRISMA-ScR could be found in S2 File. The search strategy was developed using the phrases and keywords "psychological interventions related to depression", "children and adolescent populations", and "cross-cultural adaptation". These terms were connected using Boolean operators and truncations where appropriate.

**Study eligibility criteria.** *Inclusion and exclusion criteria.* To be included in the review, papers had to report on the cross-cultural adaptation or cultural sensitivity of psychological interventions (e.g. Cognitive Behavioural Therapy (CBT), Behavioural Activation (BA), Interpersonal Therapy (IPT)) for the treatment of depression and/or anxiety disorders in young people that included the age group between 8 to 18 years. Papers describing a cross-cultural adaptation process and reporting on effectiveness were included. No restrictions were placed upon the intervention delivered, country, language, study design, year of publication or setting (e.g. school, home, community, hospital, clinic). Grey literature and conference abstracts were excluded.

**Study selection.** All the titles and abstracts were double-screened by three reviewers (MPM, LT, PK) against the defined eligibility criteria. If at any point it was unclear whether the study met the eligibility criteria, it was retained for full text screening.

Full text screening was performed by the same three reviewers, with each study independently screened by two reviewers. Any disagreements as to inclusion were discussed with a fourth reviewer (LG) until a consensus was reached. All data were managed using Rayyan software [35].

**Data extraction.** Data extraction for each study was performed independently by three reviewers (MPM, LT, PK), using a pre-piloted template. Across all studies the data extracted included: study characteristics (author(s), year of publication, study location, setting), study design, study populations (e.g. participants demographics, sample size), intervention details and comparators (including intervention type, frequency, duration) and any relevant outcome data (depression and/or anxiety). In addition, any data reporting on the methods of

intervention adaptation were extracted using Intervention Taxonomy adaptation (ITAX) templates [36].

**Quality assessment.** The methodological quality of included studies was not assessed as risk of bias assessment across studies is not applicable in scoping reviews [34].

**Data synthesis.** A narrative synthesis of all included studies was undertaken with the effectiveness of each study reported by study design and condition (anxiety, depression or both) included.

## Results

### Study selection

Overall, 2481 records were identified through electronic database searching and 1032 duplicates were removed. Title and abstract screening were undertaken for the remaining 1449 records. Following a 95% agreement rate in the 100% double-screened title and abstracts, 35 full-text articles were assessed for eligibility by the three reviewers. After relevance checking 17 studies were included in the review (Fig 1).

### Excluded studies

As shown in Fig 1, of the 35 full-text articles assessed for eligibility, 18 were excluded. Reasons for exclusion included: having the wrong population (age group more than 18 years old) (n = 9), wrong publication type (conference or dissertation abstract, review, report, editorial, study protocol) (n = 1), not mentioning cultural adaptation/related process/effectiveness (n = 7), and unobtainable full text (n = 1). As we wanted to gather evidence from the published articles, grey literature was not included.

### Included studies

**Study design.** Seventeen studies, published between 1999 and 2022 met the inclusion criteria. Of these, 11 adopted an RCT methodology [37–47], four a non-randomised trial with a single treatment arm [48–51] and the remaining 2 were case studies [52, 53]. The characteristics of included studies are presented in Table 1.

**Sample size.** The sample sizes of included studies ranged from 1 to 186 with an overall sample across included studies of n = 974. When examining these relative to study design these ranged from 18 to 186 in the RCTs and 7 to 119 in the non-randomised, single arm designs. Both case studies had a sample size of n = 1.

**Study setting.** The included studies were undertaken within 11 different countries including Puerto Rico [37, 40, 41, 53], the United States of America (USA) [39, 49, 51, 52], Turkey [48], the Philippines [42], Jordan [38], Japan [43], Pakistan [44], Spain [50], Mauritius [46], Malaysia [47] and China [45]. Most studies were conducted within clinic-based (n = 4) [37, 38, 43, 52] or school-based settings (n = 9) [39–42, 45, 47, 49–51]. One study was conducted in a university-setting [48] and two in residential care institutions [44, 46]. The remaining study [53] did not specify the setting.

**Study participants.** Across studies, participants ranged in age from 8 to 18 years. Most studies (n = 10) had a higher proportion of female than male participants with only one study [45] having a higher male than female sample. Three studies only included male participants [38, 44, 53] whilst one included only females [52], and one had equal numbers of males and females. The remaining study did not provide any information relating to the gender of participants.

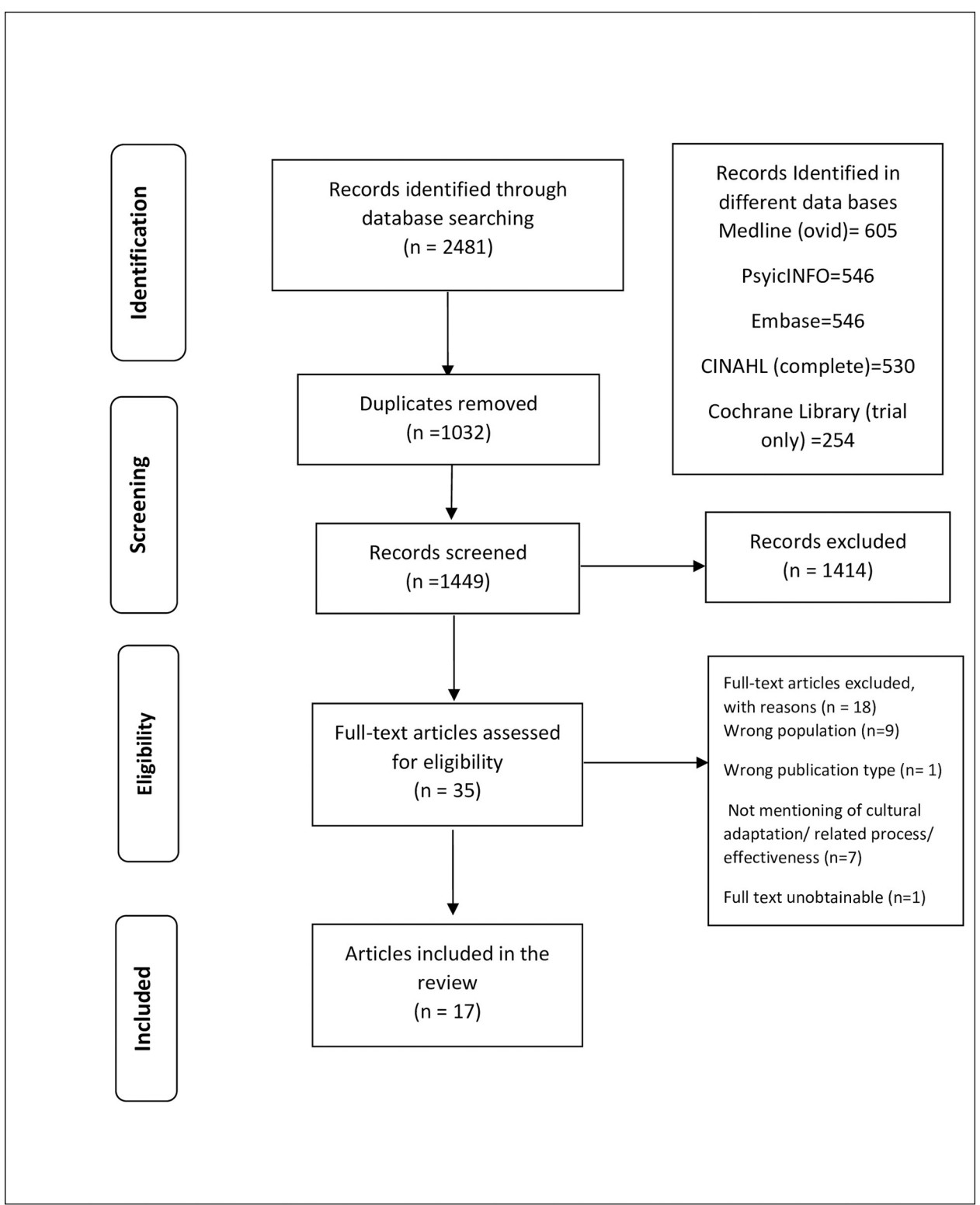

**Fig 1. PRISMA flow diagram.**

**Table 1. Characteristics of the included studies.**

| Author (year) Study design | Setting and Sample | Intervention | Comparator | Outcome Measures |
|---|---|---|---|---|
| | | **Randomised controlled trials** | | |
| Alampay et al. (2020) Pilot RCT | **Setting:** School-based, Philippines **Participants:** n = 186 **Age:** 9–16 (*M*: 11.88; *SD*: 1.84) **Gender:** 77 males (41.4%) and 109 females (58.6%) **Diagnosis:** Experience of some degree of behavioural problem, symptoms of emotional difficulties and/or peer problems (determined using SDQ) | Mindfulness-based cognitive therapy (MBCT) adapted to Kamalyan Curriculum **Duration:** 8 weekly sessions (75 minutes for younger participants, 90 minutes for older participants) **Delivery:** Group format; delivered by school guidance counsellors and teachers **n** = 87 (17 lost to follow-up) | Handicrafts (Craft based therapy) **Duration:** 8 weekly sessions (75 minutes for younger participants, 90 minutes for older participants) **Delivery:** Group format; delivered by facilitators from a women's organisation that supports livelihood activities **n** = 99 (23 lost to follow-up) | **Depression Measures:** SMFQ **Anxiety Measures:** STAIC **Timings:** End of treatment 8-weeks post-treatment |
| Bernal et al. (2019) RCT | **Setting:** Clinic-based, Puerto Rico **Participants:** n = 121 **Age:** 13–17.5 (*M, SD: NR*) **Gender:** 56 males (46.3%) and 65 females (53.7%) **Diagnosis:** CDI score ≥20, CDRS-R ≥40, met full DSM-IV criteria for MDD, maintained clinically significant depressive symptoms for at least 6 weeks before randomisation | **Name:** Culturally adapted Cognitive Behavioural Therapy and parent psychoeducation intervention (TEPSI) **Duration:** CBT: 12 sessions TEPSI: 8 2-hour sessions **Delivery:** CBT: delivered individually by Clinical Psychology graduate students TEPSI: delivered in a group format by 2 licensed Clinical Psychologists **n** = 60 (n = 13 lost to follow-up) | Culturally adapted Cognitive Behavioural Therapy **Duration:** 12 sessions **Delivery:** delivered individually by Clinical Psychology graduate students **n** = 61 (n = 14 lost to follow-up) | **Depression Measures:** CDI CDRS-R SIQ-Jr DISC-IV BDI-R **Anxiety Measures:** N/A **Timings:** Weeks 3, 5, 7, 9 of treatment End of treatment 3-months post-treatment 6-months post-treatment 9-months post-treatment 12-months post-treatment |
| Damra, Nassar & Ghabri (2014) Feasibility RCT | **Setting:** Clinic-based, Jordan **Participants:** n = 18 **Age:** 10–12 (*M* = 11.3; SD:NR) **Gender:** All-male sample **Diagnosis:** Suffering from physical abuse and clinical symptoms of PTSD and depression for at least five weeks prior to treatment | Trauma-Focused Cognitive Behavioural Therapy (TF-CBT) adapted for the Jordanian culture **Duration:** 10 60-minute sessions delivered over 2 weeks **Delivery:** Group-format delivered by two accredited children's counsellors **n** = 9 (0 lost to follow-up) | Wait-list control **n** = 9 | **Depression Measures:** CDI **Anxiety Measures:** N/A **Timings:** End of treatment 4-months post-treatment |
| Ishikawa et al. (2019) RCT | **Setting:** Clinic-based, Japan **Participants:** n = 51 **Age:** 8–15 (*M* = 10.90; *SD* = 2.00) **Gender:** 22 males (43.1%), 29 females (56.9%) **Diagnosis:** Experiencing an anxiety disorder as determined using the ADIS | The Japanese Anxiety children/ Adolescents Cognitive Behavioural Therapy (JACA-CBT) **Duration:** 8 1-hour weekly sessions (with an optional 3 booster sessions) **Delivery:** Face-to face delivered by 2 clinical psychologists **n** = 26 | Wait-list control n = 25 | **Depression Measures:** CDI DSRS **Anxiety Measures:** ADIS SCAS-C/SCAS-P **Timings:** Pre-treatment End of treatment 3-months post-treatment 6-months post-treatment |
| Khan, Malik, Ahmed & Riaz (2020) Feasibility RCT | **Setting:** Orphanage, Pakistan **Participants:** n = 24 **Age:** 8–13 (*M, SD: NR*) **Gender:** All males Diagnosis: >30 on SCARED, not currently taking any medication nor receiving psychotherapy | Coping Cat (CC) Cognitive Behavioural Therapy adapted into Urdu **Duration:** 12 40–60 minute sessions delivered weekly **Delivery:** Face-to-face delivered by post-graduate clinical psychologist **n** = 12 | Wait-list control **n** = 12 (9 lost to follow-up) | **Depression Measures:** N/A **Anxiety Measures:** SCARED CASI **Timings:** 1-week pre treatment Post-session 3 End of treatment 6-months post-treatment |

*(Continued)*

**Table 1.** (Continued)

| Author (year) Study design | Setting and Sample | Intervention | Comparator | Outcome Measures |
|---|---|---|---|---|
| Li et al. (2022) Pilot RCT | **Setting:** Two primary schools, China <br> **Participants:** n = 87 <br> **Age:** 8–12 (M = 11, SD = .75) <br> **Gender:** 54 male (62%), 33 female (38%) <br> **Diagnosis:** Experience of at least one traumatic event and meets DSM–5 PTSD criterion for PTSD or partial PTSD using PCL-5 checklist | Adapted version of Cohen's trauma-focused cognitive behavioural therapy (TF-CBT) <br> **Duration:** 10-sessions, each lasting up to 50 minutes <br> **Delivery:** Face-to-face; 7 group sessions and 3 individual sessions, delivered by members of the protocol developer's team (counsellors) from Beijing Normal University who held master's degrees in social work <br> n = 45 | Treatment as usual at school site—usually including emotional catharsis techniques <br> **Duration:** number of sessions decided by young person and school psychologist, each session lasting c. 45 minutes <br> **Delivery:** School psychologists n = 42 | Depression Measures: CDI-S <br> Anxiety Measures: CSAS-C <br> Timings: <br> Baseline <br> 2-weeks post-treatment <br> 3-months post-treatment |
| Listug-Lunde et al. (2013) Pilot RCT | **Setting:** One rural northern plains tribal school, USA <br> **Participants:** n = 19 <br> **Age:** 11–14 (CWD-A group: 11–14 years, M = 12.38, SD = .92, TAU group: 12–14 years, M = 12.5, SD = 1.07) <br> **Gender:** 10 males (62.5%), 6 females (37.5%) (remaining 3 unknown) <br> **Diagnosis:** Moderate levels of depression defined by raw scores of 15 or higher on the CDI | Skills Development Class (culturally adapted Coping with Depression Course for Adolescents: CWD-A) <br> **Duration:** 13 35 to 40-minute sessions, bi-weekly for 7 weeks followed by 2 booster sessions held within 1-month post-intervention <br> **Delivery:** Group-format delivered by two therapists: One Indian Health Services mental health professional and one graduate student <br> **n** = 10 (2 lost to follow-up) | Treatment as usual <br> **Duration:** NR <br> **Delivery:** delivered by two therapists: One Indian Health Services mental health professional and one graduate student <br> **n** = 9 (1 lost to follow-up) | **Depression Measures:** CDI <br> **Anxiety Measures** MASC <br> **Timings:** <br> Pre-treatment <br> Post-treatment <br> 3-months post-treatment |
| Ramdhonee-Dowlott al. (2021) RCT | **Setting:** 6 residential care institutions (RSI), Mauritius <br> **Participants:** n = 100 <br> **Age:** 9–14 (M = 11.75, SD = 1.97) <br> **Gender:** 24 males (24%), 76 females (76%) <br> **Diagnosis:** RCI report on child/adolescent displaying difficulties in managing emotions or emotional distress, no clinical cut-off but all scored above 5 on the Emotional Symptoms subscale of the self-report SDQ | Super Skills for Life (SSL) manualised CBT translated into French and adapted for Mauritian culture and use in RCI settings <br> **Duration:** 8 weekly sessions of c.45 minutes <br> **Delivery:** Face-to-face group format, delivered by developmental psychologist with 5 years of experience, assisted by one staff member at each host RCI <br> n = 50 | Wait-list control <br> n = 50 | **Depression and Anxiety Measures:** RCADS <br> Video speech tasks <br> **Timings:** <br> 1-week pre-treatment <br> 1-week post-treatment <br> 3-months post-treatment |
| Rosselló & Bernal (1999) RCT (3 arm) | **Setting:** School-based, Puerto Rico <br> **Participants:** n = 71 <br> **Age:** 13–11 (M: 14.10; SD: 1.40) <br> **Gender:** 33 males (46%) and 38 Females (54%) <br> **Diagnosis:** Scoring over 11 on the CDI and who met DSM-Il criteria for depression on the DISC-2 (parent or adolescent versions) | Cognitive Behavioural Therapy adapted for Puerto-Rican youth <br> **Duration:** 12 sessions delivered weekly <br> **Delivery:** Face-to-face, individual delivery by advanced graduate clinical psychology students <br> **n** = 25 <br> Interpersonal Therapy (IPT) adapted for Puerto-Rican youth <br> **Duration:** 12 sessions delivered weekly <br> **Delivery:** Face-to-face individual delivery by advanced graduate clinical psychology students <br> **n** = 23 | Wait-list control <br> **n** = 23 | **Depression Measures:** CDI <br> **Anxiety Measures:** N/A <br> **Timings:** <br> Pre-treatment <br> End of treatment <br> 3-months post-treatment |

*(Continued)*

**Table 1.** (Continued)

| Author (year) Study design | Setting and Sample | Intervention | Comparator | Outcome Measures |
|---|---|---|---|---|
| Rosselló, Bernal & Rivera-Medina (2008) RCT (4 arms) | **Setting:** Schools, Puerto Rico **Participants:** n = 112 **Age:** 12–18 (*M*: 14.52; *SD*: 1.85). **Gender:** 50 males (44.6%) and 62 Females (55.4%) **Diagnosis:** Scoring over 11 on the CDI and who met DSM-Il criteria for depression on the DISC-2 (parent or adolescent versions) | Individual Cognitive Behavioural Therapy **Duration:** 12 sessions delivered weekly **Delivery:** Face-to-face individual delivery by advanced graduate clinical psychology students **n** = 23 Individual Interpersonal Psychotherapy **Duration:** 12 sessions delivered weekly **Delivery:** Face-to-face individual delivery by advanced graduate clinical psychology students **n** = 31 | Group Cognitive Behavioural Therapy **Duration:** 12 sessions delivered weekly **Delivery:** Face-to-face group-format by advanced graduate clinical psychology students **n** = 29 Group Interpersonal Psychotherapy **Duration:** 12 sessions delivered weekly **Delivery:** Face-to-face individual delivery by advanced graduate clinical psychology students **n** = 29 | **Depression Measures:** CDI **Anxiety Measures:** N/A **Timings:** Pre-treatment End of treatment 3-months post-treatment |
| Saw, Tam & Bonn (2019) Pilot RCT | **Setting:** School, Malaysia **Participants:** n = 20 **Age:** 16 (*M*, *SD*: NR) **Gender:** 10 males (50%) and 10 females (50%) **Diagnosis:** raw score above the cut-off point of 76 on RADS-2 | Group Cognitive Behavioural Therapy **Duration:** 8 ninety-minute sessions delivered weekly **Delivery:** Group format, delivered by trained local school counsellors **n** = 10 | Wait-list control **n** = 10 | **Depression Measures:** RADS-2 ATQ-Malay **Anxiety Measures:** N/A **Timings:** Pre-treatment Mid-treatment (after session 4) Post-treatment (after session 8) 1-month post-intervention |
| | | **Non-randomised trials (single treatment arm)** | | |
| Acarturk et al. (2019) Pilot study | **Setting:** University-based, Turkey **Participants:** n = 13 **Age:** 13–17 (*M*: 14.9, *SD*: NR) **Gender:** 4 males (30.8%), 9 females (69.2%) **Diagnosis:** Diagnosis of anxiety or mood disorder based on K-SADS-PL despite taking a maximally tolerated SSRI for at least 12 weeks | Culturally Adapted Transdiagnostic Cognitive Behavioural Therapy (CA-CBT) **Duration:** 10 1-hour sessions **Delivery:** Group format, delivered by two psychologists. **n** = 13 (0 lost to follow-up) | Not applicable | **Depression Measures:** BDI **Anxiety Measures:** SCARED **Timings:** 1-week pre-treatment 1-week post-treatment 2-months post-treatment |
| Goodkind, LaNoue & Milford (2010) Pilot study | **Setting:** School-based, USA **Participants:** n = 24 **Age:** 12–15 (*M*: 13.39, *SD*: NR) **Gender:** 7 males, 16 females **Diagnosis:** Past trauma experience. Specific screening criteria and cut-off scores unclear | Cognitive Behavioural Intervention for Trauma in Schools (CBITS) **Duration:** 10 weekly sessions **Delivery:** Group-format delivered by a member of clinical staff and a cofacilitator **n** = 23 (1 lost to follow-up) | Not applicable | **Depression Measures:** CDI **Anxiety Measures:** MASC **Timings:** Pre-treatment End of treatment 3-months post-treatment 6-months post-treatment |
| Morsette et al. (2009) | **Setting:** Rural American Indian Reservation, USA **Participants:** n = 7 (only 4 completed) **Age:** 11 to 12 years (*M*, *SD*: NR) **Gender:** *NR* **Diagnosis:** Posttraumatic Stress Disorder (PTSD) and symptoms of depression with history of trauma | Culturally adapted Cognitive Behavioral Intervention for Trauma in Schools (CBITS) **Duration:** 10-week course of CBITS—Cognitive Behavioral Intervention for Trauma in Schools **Delivery:** Group-format **n** = 7 (3 lost to follow-up) | Not applicable | **Depression Measures:** CDI **Anxiety Measures:** CPSS **Timings:** Baseline Pre-treatment Post-treatment |

*(Continued)*

**Table 1.** (Continued)

| Author (year) Study design | Setting and Sample | Intervention | Comparator | Outcome Measures |
|---|---|---|---|---|
| Orgilés, Fernández-Martínez, Espada & Morales (2019) One-group quasi-experimental design | **Setting:** School-based, Spain **Participants: n** = 119 **Age:** 8–12 (*M* = 9.39; *SD* = 1.26) **Gender:** 68 males (57.1%) and 51 females (42.9%), **Diagnosis:** ≥4 on emotional symptoms subscale of the SDQ–parent version | Spanish adapted super skills for life (SSL) intervention for the prevention of anxiety and depression **Duration:** 8 45-minute weekly sessions over 8 weeks **Delivery:** Group-format delivered by 6 psychologists **n** = 119 (9 lost to follow-up) | Not applicable | **Depression Measures:** CDI **Anxiety Measures:** SCARED CALIS-C **Timings:** Pre-treatment End of treatment 1-year post-treatment |
| | | **Case studies** | | |
| Binkley & Koslofsky (2016) | **Setting:** Clinic-based, USA **Participants:** n = 1 **Age:** 17 **Gender:** Female **Diagnosis:** Diagnosis of moderate MDD with comorbid bulimia nervosa | Family-Based Therapy for Bulimia Nervosa (FBT-BN) **Duration**: 5 sessions over 3 months **Delivery:** Delivered face-to-face, information about delivering clinician *NR* **n** = 1 | Not applicable | **Depression Measures:** CDI **Anxiety Measures:** N/A **Timings:** 1-month post-treatment |
| Duarté -Vélez, Bernal & Bonilla (2010) (Case study within larger RCT) | **Setting:** *NR, Puerto Rico* **Participants:** n = 1 **Age:** 16 years **Gender:** Male **Diagnosis:** Major Depressive Disorder (MDD), non-specified anxiety disorder and Attention Deficit and Hyperactivity Disorder (ADHD) | Culturally adapted manual-based Cognitive Behavioural Therapy **Duration:** 12 weekly sessions (with 4 optional additional sessions) **Delivery:** Individually delivered by a psychotherapist **n** = 1 | Not applicable | **Depression Measures:** CDI CDRS-R **Anxiety Measures:** N/A **Timings:** Pre-treatment Session-by-session End of treatment 6-months post-treatment 9-months post-treatment 12-months post-treatment 15-months post-treatment |

**Notes.** *NR*: Not reported. *Depression Measures*: ATQ-Malay: Automatic Thoughts Questionnaire—Malay version (Oei & Mukhtar, 2008); BDI: Beck Depression Inventory (Beck, Steer & Brown, 1996); BDI-R: Beck Depression Inventory Revised (Bonilla, Bernal, Santos & Santos, 2004); CDI: Children's depression Inventory (Kovacs, 1985; 2011); CDI-S: Children's Depression Inventory Short Form (Vega et al., 2016); CDRS-R: Children's depression rating scale–revised (Poznanski & Mokros, 1996); DSRS: Depression Self-Rating Scale (Birleson, 1981); K-SADS-PL: Kiddie Schedule for affective disorders present and lifetime version (Kaufman et al., 1997); SIQ-JR: Suicidal Ideation Questionnaire-Junior (Reynolds, 1988); RADS-2: Reynolds Adolescent Depression Scale–Second Edition (Reynolds, 2002); SMFQ: Short Mood and Feelings Questionnaire (Angold et al., 1995). *Anxiety Measures*: ADIS: The Anxiety Disorders Interview Schedule (Silveman & Albano, 1996); CALIS-C: Child Anxiety Life Interference Scale (Lyneham et al., 2013); CASI: Childhood Anxiety Sensitivity Index (Silverman et al., 1991); CSAS-C: Chinese version of the State Anxiety Scale for Children (Li & Lopez, 2007); CPSS: Child PTSD Symptom Scale (Foa et al., 2001); MASC: Multi-dimensional anxiety scale for children (March & Parker, 1999); SCARED: Screen for anxiety related emotional disorders (Birmaher et al., 1999); SCAS: C: Spence Children's Anxiety Scale Child versions (Spence, 1998); SCAS: P: Spence Children's Anxiety Scale Parent versions (Nauta 2004); STAIC: State-Trait Anxiety Inventory for Children (Speilberger, 1973). Depression and Anxiety Measures: RCADS: The Revised Children's Anxiety and Depression Scale (Chorpita et al., 2000)

All studies focused upon interventions designed to treat anxiety [43, 44], depression [37, 40, 41, 47, 53], both anxiety and depression [39, 42, 46, 48, 50] or depression alongside another co-morbid condition including an eating disorder [52], Post-Traumatic Stress Disorder (PTSD) [38], or both PTSD and anxiety [45, 49, 51].

**Interventions and comparators.** The psychological interventions included, CBT (53), mindfulness-based cognitive therapy (MBCT) [42]; CBT [40, 41, 43, 44], transdiagnostic CBT [46–48], adding a parent psychoeducation intervention (TEPSI) as part of CBT [37], Trauma-Focused CBT (TF-CBT) [38, 45], Cognitive Behavioural Intervention for Trauma in Schools (CBITS) [49, 51], a transdiagnostic protocol based on CBT designed for children with internalizing problems [50], a culturally adapted version of the coping with depression course for

adolescents (CWD-A) called 'Skills Development Class' [39] and Family-Based Therapy for Bulimia Nervosa (FBT-BN) [52].

Three studies reported interventions directly involving parents [37, 52, 53]. One of these, an RCT, added a parent psychoeducation intervention (TEPSI) as part of CBT for adolescents with Major Depressive Disorder (MDD) and compared that with CBT alone [37]. The remaining studies reported interventions targeted to adolescents only.

All studies were delivered face-to-face with some delivered individually and some (n = 9) in groups. In one study [45] young people participated in both group and individual sessions. For the studies involving parents, one of these incorporated separate and concurrent psychoeducation sessions with the parents [37]; one a joint session with the participant's mother in addition to individual face-to-face meetings between the mother and the professional [53] and one with sessions attended by the child and both parents [52].

**Cross-cultural adaptation process.** Some studies described the underlying framework used for the cross-cultural adaptation of their intervention, while others simply mentioned that their intervention was cross-culturally adapted without referring to any framework/ model. The cross-cultural adaptation frameworks used in the included studies are presented in Fig 2.

Six of the seventeen included studies [37, 40, 41, 47, 52, 53] used the ecological validity and cultural sensitivity framework [25] for the cultural adaptation of the intervention that they delivered. Here eight elements (language, person, metaphors, content, concept, goals, methods, context) are considered in the adaptation process. Four of these studies involved Puerto Rican adolescents [37, 40, 41, 53]. In the pilot RCT by Saw, Tam and Bonn [47], the treatment module which was designed for use in Puerto Rico [54] was transformed into an intervention 'Shine Through Any Roadblocks' (STAR) suitable for the Malaysian high school students.

Goodkind and colleagues [49] described adapting their intervention at both surface and deep structure levels to make it culturally sensitive [26, 27]. The surface structure considerations are those which affect the "fit" of the intervention to the context. For example, a change of materials, channels of delivery, or settings constitute surface structure changes, which would result in a culturally targeted intervention. Deep structural changes consider cultural, social, psychological, environmental, and historical factors, which may be unique to a particular racial or ethnic group. The adaptation process included some deep structural changes, for instance using stories and examples based on participants' cultural beliefs, as well as more surface structure changes to determine whether an existing intervention designed specifically for youth of diverse backgrounds was feasible, acceptable, appropriate, and effective for American Indian youth. There is an interesting question here of how far it is necessary to probe into the deep structural elements—the cultural, social, psychological, environmental, and historical factors–to have the requisite knowledge and understanding to properly shape the more 'surface' adjustments, for instance deciding where the intervention is delivered or how much parents are involved.

Ramdhonee-Dowlot and colleagues [46] used a predominantly surface-level approach for adaptation. Two bilingual psychologists translated the original manualised intervention from English to French. An expert focus group discussion was carried out with five professionals working with children in residential care institutions (RCIs) and the programme was then pilot-tested with eight children aged 9–14 years. Following this, slight modifications (including to contexts, language expressions and examples) were implemented to make the intervention more appropriate to both the Mauritian culture and the specific population of children in RCIs.

Khan and colleagues [44] used the planned adaptation approach [28] to improve the cultural fit of the intervention while maintaining the core elements. They translated and adapted

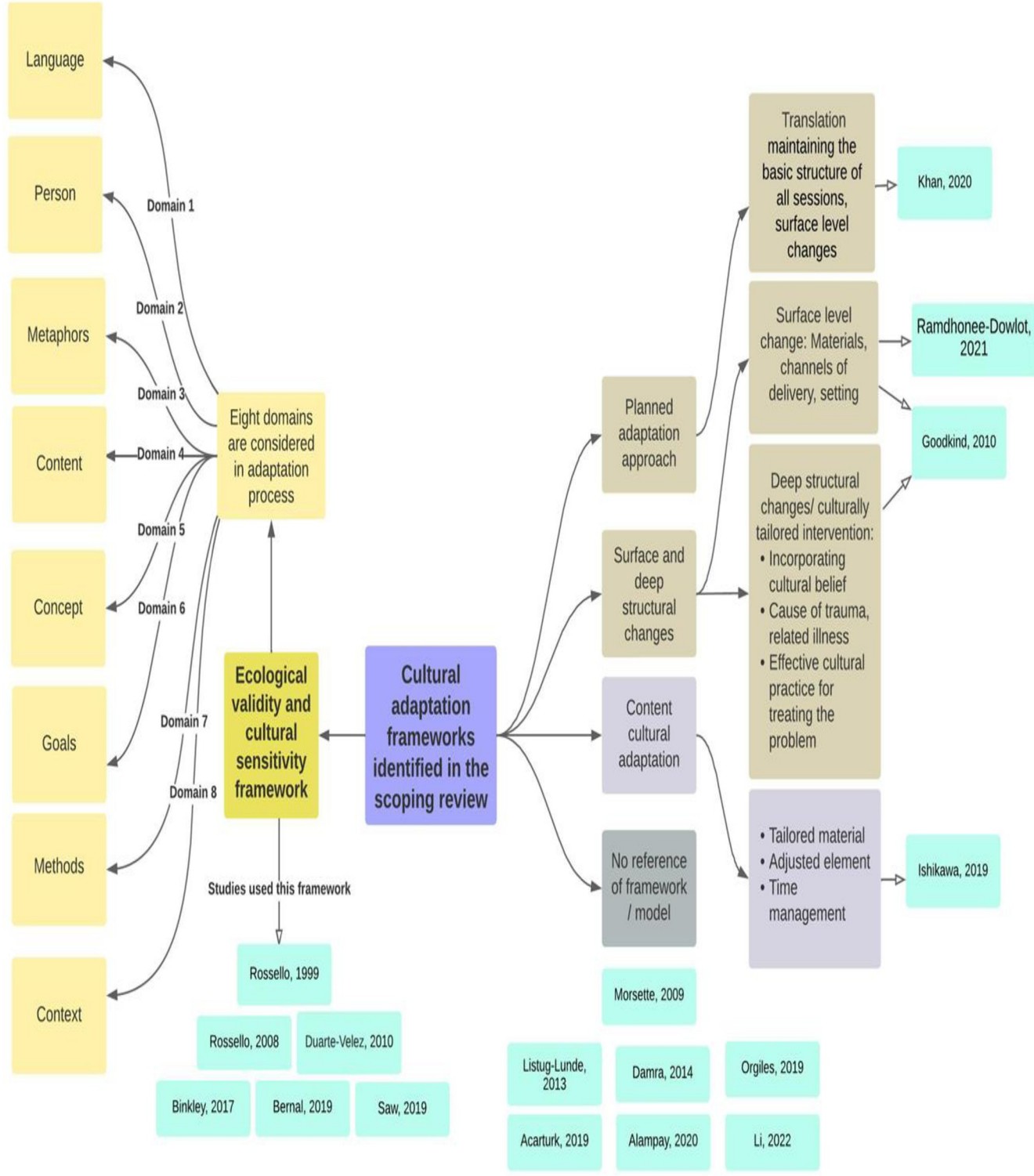

**Fig 2. Cross-cultural adaptation frameworks used in the included studies.**

the workbook associated with their CBT-based intervention into Urdu. Special consideration was given to translating the emotional expressions, cultural jargons, and acronyms. Culturally specific information considered the developmental perspectives of Pakistani children. Some of the pictures, and names of characters were changed so that children could relate more easily.

Ishikawa and colleagues [43] described 'content cultural adaptation', a bidirectional approach to cultural adaptation drawing on the work of others [29, 30], to increase the suitability for Japanese children. Their treatment was adapted over four phases. Each adaptation followed pilot testing of the program, allowing feedback from therapists to be incorporated into the next iteration. Cultural adaptations and modifications of both context and content were made in the Japanese Anxiety Children/Adolescents CBT program (JACA-CBT).

Some studies discussed the details of the adaptation process without referring to an underlying theory [39, 51]. Morsette and colleagues [51] in their study included significant attempts to reshape the intervention to match a different set of cultural circumstances and values but did so without (making explicit) reference to a pre-established framework. American Indian health professionals, elders or spiritual leaders, teachers, and counsellors were consulted in modifying training case examples for students, to add native linguistic concepts, and to embed local history and allegories in the lessons.

Listug-Lunde and colleagues [39] delivered the modified CWD-A course [55] with rural American Indian middle school students. The modification was conducted in consultation with educators, school and community mental health professionals, and an expert in American Indian mental health issues. Additional modifications for cultural sensitivity and relevance were made, including offering the intervention as part of the regular school schedule (class credit was provided), changing examples and role-play situations to reflect culturally appropriate and relevant activities, and adding discussions about the cultural impact of skills such as assertiveness, eye contact, constructive criticism, and self-disclosure.

Some studies just briefly mentioned that their intervention was cross-culturally adapted without discussing either the theory or details of the adaptation process [38, 42, 45, 48, 50]. Acarturk and colleagues [48] mentioned that their intervention treatment manual was culturally adapted by emphasising somatic experiences and eliciting cultural ideas about symptoms. The main components of the transdiagnostic CBT were kept while cultural elements such as metaphors were added to the intervention. In Alampay et al. [42] the manualized Kamalyan Curriculum was based on adaptations of Segal et. al's MBCT program [56], MBCT for children [28, 57], and exercises from Sitting Still Like a Frog [58, 59]. Orgilés and colleagues [50] described a Spanish-adapted version of the transdiagnostic protocol Super Skills for Life (SSL) [60] in a community sample of Spanish-speaking children aged 8–12 years with symptoms of anxiety and/or depression. Li et al. [45] tested an intervention that was developed based on core elements of the authoritative version of Cohen's TF-CBT therapy [61] with the specific content of each session modified or designed by researchers. Significant modifications were also made to cater to Chinese children and implementation contexts.

A community-based participatory approach was used for the cross-cultural adaptation process in two of the included studies. Goodkind and colleagues [49] described that the intervention was developed, implemented, and evaluated in partnership with three American Indian communities that emphasised traditional cultural teachings, parenting and social skills building, healing historical trauma, and cultural practices e.g. equine activities. Another study conducted with American Indian young people by Morsette et al. [51] equally involved community elders in adapting the intervention, recognising the pivotal role they hold and the respect afforded them in these communities. In an effort to further assimilate and increase the cultural validity of the intervention, the elders were present for sessions at the start and the end that included prayers and other cultural practices. among the included studies.

**Narrative synthesis of the effectiveness of the interventions.** Information relating to the effectiveness of the interventions reported in the included papers are presented relative to the study methodology employed. Table 2 depicts the methodology employed as well as whether a study examined depression and anxiety combined, depression alone, or anxiety alone and the effectiveness of the intervention as reported in the paper. Whilst all the included studies reported secondary outcomes, the focus here is only on results relating to depression and/or anxiety.

Overall, the results indicate potential effectiveness of cross-culturally adapted interventions. Apart from the pilot study by Alampay et al. (2020) [42], all the studies reported that the culturally adapted interventions resulted in improvements in depression and/or anxiety symptoms. However, most of the studies employed small sample sizes and were not powered to test the statistical significance of effectiveness. Due to the heterogeneity of the studies and use of different outcome measures, it was not possible to combine the results. Therefore, we reported the effectiveness of the interventions across studies individually.

*Randomised controlled trials.* The majority of included studies (n = 11) adopted an RCT methodology. Of these, five examined the delivery of a culturally adapted intervention with young people experiencing both anxiety and depression [39, 42, 43, 45, 46], five with depression alone [37, 38, 40, 41, 47], and one focusing upon anxiety alone [44] (Table 2).

*Anxiety and depression combined.* In the study by Alampay and colleagues [42], young people were randomised to receive either a mindfulness-based cognitive therapy (MBCT) adapted to Kamalyan Curriculum or Handicrafts (a craft -based therapy). Depression scores, as measured by the SMFQ were seen to reduce over time for young people randomised to the Handicrafts group (baseline: $M$ = 0.72, $SD$ = 0.34, post-intervention: $M$ = 0.65, $SD$ = 0.36; 2 months follow-up: $M$ = 0.53, $SD$ = 0.35). However, this was not the same for the MBCT group, where depression scores remained stable (baseline: $M$ = 0.74, $SD$ = 0.32, post-intervention: $M$ = 0.77, $SD$ = 0.39, 2 months follow-up: $M$ = 0.68, $SD$ = 0.37). For anxiety, scores as measured by the STAIC remained stable across time-points for both the Handicrafts (baseline: $M$ = 1.99, $SD$ = 0.33, post-intervention: $M$ = 1.96, $SD$ = 0.37; 2 months follow-up: $M$ = 1.97, $SD$ = 0.31) and MBCT group (baseline: $M$ = 2.11, $SD$ = 0.26, post-intervention: $M$ = 2.08, $SD$ = 0.29, 2 months follow-up: $M$ = 1.99, $SD$ = 0.32). The authors concluded that whilst those in the Handicrafts group demonstrated decreases in depression scores over time, those randomised to the MBCT group did not see any changes to their depression or anxiety.

In the study by Ishikawa et al [43] which compared culturally adapted CBT (the Japanese Anxiety Children/Adolescents Cognitive Behavior Therapy program: JACA-CBT) to a wait-list control, a significant difference was found between groups regarding the number of participants free of their principal diagnosis (anxiety disorder) at post-treatment. 50% (n = 13) of those randomised to CBT were free of their principal diagnosis at post-treatment in comparison to 12% (n = 3) of those in the wait-list control ($\chi2$ (1, N = 51) = 8.55, $\eta2$ = 0.17, $p$< .01). Those randomised to the CBT group showed significant improvements in clinical severity rating (CRS) scores on the Anxiety Disorders Interview Schedule for DSM-IV (ADIS) from pre- to post-treatment (Pre: $M$ = 6.31, $SE$ = 0.49, Post: $M$ = 3.08, $SE$ = 0.50) in comparison to those in the wait-list control (Pre: $M$ = 6.72, $SE$ = 0.50; Post: $M$ = 6.00, $SE$ = 0.51). In terms of depression, only those randomised to CBT demonstrated statistically significant reductions in scores as measured by the DSRS and CDI from pre- to post-treatment.

Ramdhonee-Dowlet, Balloo & Essau (2021) [46] randomised young people to either the Super Skills for Life (SSL) intervention or a wait-list control. Across all five anxiety subscales of the RCADS (General Anxiety Disorder, Separation Anxiety Disorder, Panic Disorder, Social Phobia, Obsessive compulsive Disorder), those in the intervention group reported significantly lower scores at both post-intervention and 3-months follow-up than those in the wait-list

**Table 2. Included study methodologies, condition and effectiveness of the interventions.**

| Study Design | Study | Depression Only | Anxiety Only | Depression and Anxiety | Effectiveness of the interventions (as reported by the studies) |
|---|---|---|---|---|---|
| Randomised Controlled Trials | Alampay et al. (2020) | | | √ | Participation in the Kamalayan program did not affect depression or anxiety. |
| | Bernal et al. (2019) | √ | | | The culturally adapted cognitive-behavioral therapy (CBT) was found to be effective with Latino/a adolescents showing clinically significant improvements from pretreatment to posttreatment and remained stable at a 1-year follow-up. |
| | Damra, Nassar & Ghabri (2014) | √ | | | Significant post-treatment improvements for the Trauma-Focused CBT (TF-CBT) group in all outcome measures and sustainability of the treatment gains for the TF-CBT group at 4-months follow-up. |
| | Ishikawa et al. (2019) | | | √ | Findings support the transportability of CBT and the efficacy of a bidirectional, culturally adapted CBT in an underrepresented population. |
| | Khan, Malik, Ahmed & Riaz (2020) | | √ | | Intent-to-Treat analysis indicated significant decreases in self-reported anxiety sensitivity; anxiety; generalized anxiety disorder and panic disorder in the treatment group from pre- to follow-up compared to the control group. |
| | Li et al. (2022) | | | √ | Little change found from posttreatment to 3-month follow-up. The findings indicated that the school-based group CBT Power up Children's Psychological Immunity (PCPI) intervention was feasible and acceptable. Further evaluation is needed to examine its effectiveness in a study employing a larger sample size. |
| | Listug-Lunde et al. (2013) | | | √ | Findings suggest the Adolescent Coping with Depression (CWD-A) is a promising approach for reducing depressive and anxiety symptoms in rural American Indian students and should be further evaluated with a larger sample of students |
| | Rosselló & Bernal (1999) | √ | | | Results suggest that interpersonal psychotherapy (IPT) and CBT significantly reduced depressive symptoms compared to a wait-list control. |
| | Rosselló, Bernal & Rivera-Medina (2008) | √ | | | Findings suggest that CBT and IPT are robust treatments in both group and individual formats. However, CBT produced significantly greater decreases in depressive symptoms and improved self-concept than IPT. |
| | Ramdhonee-Dowlet, Balloo & Essau (2021) | | | √ | Findings provide evidence for the effectiveness of a transdiagnostic prevention programme for emotional problems in residential care institutions in a low- and middle-income country |
| | Saw, Tam & Bonn (2019) | √ | | | Findings indicate that the Malay-language 'STAR' CBT protocol could be an effective means of reducing depressive symptoms among Malaysian high school students in school settings. |
| Non-Randomised Controlled Trials (Single Arm) | Acaturk et al. (2019) | | | √ | Findings suggest that culturally adapted transdiagnostic CBT (CA-CBT) is effective in reducing anxiety and depression symptoms |
| | Goodkind, LaNoue & Milford (2010) | | | √ | Improvements in anxiety and depression were maintained 6-months post-intervention |
| | Morsette et al. (2009) | | | √ | Posttraumatic Stress Disorder (PTSD) and depressive symptoms decreased for three of the four students who completed treatment of Cognitive Behavioral Intervention for Trauma in Schools (CBITS) |
| | Orgilés, Fernández-Martínez, Espada & Morales (2019) | | | √ | Anxiety and depressive symptoms were significantly reduced at post-test and 12-month follow-up. |
| Case Studies | Binkley & Koslofsky (2017) | √ | | | By the end of treatment, there was a reduction of depressive symptoms |
| | Duarté -Vélez, Bernal & Bonilla (2010) | √ | | | Remission of depression reported following CBT. |

control group. At post-intervention total anxiety scores were $M = 36.53$, $SE = 1.47$ for those in the intervention group compared to $M = 85.79$, $SE = 1.49$ for those in the wait-list control. At the 3-month follow-up total anxiety scores were $M = 40.56$, $SE = 2.55$ for those who received the intervention compared to $M = 72.28$, $SE = 2.58$ for those in the wait-list control group. However, no significant differences between groups were found on video speech tasks which were used as behavioural assessments of anxiety. The intervention group also demonstrated lower scores on the depression subscale of the RCADS at post-intervention ($M = 6.14$, $SE = 0.39$) and 3-months follow-up ($M = 7.06$, $SE = 0.39$) in comparison to the wait-list control group (post-intervention: $M = 19.10$, $SE = 0.39$, 3-months follow-up: $M = 20.36$, $SE = 0.39$). These findings, as well as large effect sizes for all significant results, suggest that SSL led to reductions in both anxiety and depressive symptoms.

In the study by Li et al. (2022) [45] participants received either 'the Power up Children's Psychological Immunity' (PCPI) intervention adapted from Trauma Focused CBT [61] or treatment as usual (TAU). At post-treatment those in the PCPI group demonstrated significantly lower anxiety severity than those in the TAU group ($M = 17.87$, $SD = 4.17$, vs. $M = 19.99$, $SD = 4.44$, $p = 0.26$) as well as reduced PTSD scores ($M = 21.90$, $SD = 15.89$ vs. $M = 28.59$, $SD = 16.69$, $p = .048$) however no significant differences in depression severity were reported at this time-point. In addition, no significant differences were found on PTSD scores nor anxiety or depression severity between the two groups at 3-months follow-up. Despite these findings, the authors found PCPI to be feasible to deliver and reported high levels of acceptability, adherence and participant satisfaction.

Listug-Lunde and colleagues [39] compared the Skills Development Class—culturally adapted Coping with Depression course for Adolescents: CWD-A [55]—with TAU for a group of rural American Indian middle school students. The results demonstrated statistically significant reductions in depression scores on the CDI across both groups from pre- to post-treatment and to 3-months follow-up ($F (2, 28) = 10.09$, $p < .01$). Significant differences were also found between pre-treatment and post-treatment depression scores ($t(15) = 2.843$, $p < .05$) and between pre-treatment and 3-month follow-up ($t(15) = 5.256$, $p < .001$) across both groups. At baseline three participants in each treatment arm were experiencing clinically significant levels of depression symptoms but this had reduced to two participants per group at post-treatment and one participant per group at the 3-month follow-up. Although reductions were seen in anxiety scores across all three time points for the CWD-A group and from pre- to post-treatment in the TAU group, no statistically significant differences were reported.

*Depression*. Five of the included RCTs [37, 38, 40, 41, 47] examined the effectiveness of delivering a culturally adapted intervention to young people experiencing depression. Whilst a range of depression measures were used across studies all administered the Children's Depression Inventory (CDI).

In the study by Bernal et al. [37] depression symptoms, as measured by both the CDI and the CDRS-R were seen to reduce from pre- to post-treatment in both treatment groups (CBT and TEPSI, CBT alone). From pre-treatment to post-treatment, 76.7% of those randomised to CBT alone and 80.4% of those who received CBT and TEPSI had demonstrated a reliable change in depression scores. At 12months post-treatment, the percentage of those remaining in remission for MDD were: 69.5% for those in the CBT group only and 67.2% for those in the CBT and TEPSI groups. Whilst reductions in depression scores were reported across both treatment groups, the CBT and TEPSI group was not seen to be clinically more effective in reducing depression scores than CBT alone.

In the study by Damra and colleagues [38] a trauma-focused CBT (TF-CBT) was adapted for use with young people and compared to a wait-list control. Most TF-CBT components received positive feedback and were deemed acceptable by children, parents, and child

counsellors. While baseline scores on the CDI were similar between the TF-CBT and wait-list groups, those in the intervention arm demonstrated lower CDI scores at both post-treatment (TF-CBT: $M = 26.44$, $SD = 3.84$, wait-list control: $M = 43.77$, $SD = 3.84$) and at 4-months follow-up (TF-CBT: $M = 27.44$, $SD = 1.94$, wait-list control: $M = 44$, $SD = 1.87$). The differences across time points were significant ($F = 99.617$, $df = 1.247$, $p < .05$) suggesting that TF-CBT was effective in reducing symptoms of depression.

In the study by Rosselló & Bernal [40] participants were randomised to receive one of two culturally adapted treatments (CBT or IPT) or a wait-list control. At post-treatment, lower depression scores, as measured using the CDI, were found in participants who received IPT ($M = 10.79$, $SD = 6.51$) or CBT ($M = 13.28$, $SD = 7.61$) compared to the wait-list control ($M = 15.83$, $SD = 6.83$). Orthogonal comparisons demonstrated that those in both the IPT and CBT groups had statistically lower depression scores at post-treatment compared to the wait-list control but no statistically significant differences were found on the CDI scores between the IPT and CBT groups. Furthermore 77% of those randomised to IPT and 67% of those to CBT had better outcomes in relation to depressive symptoms at post-treatment than those in the wait-list control group.

In a later study, Rosselló and colleagues [41] compared culturally adapted treatments (CBT and IPT) delivered in both individual and group formats. Reductions in depression scores on the CDI were reported from pre-treatment (IPT: $M = 21.52$, $SD = 6.88$; CBT: $M = 22.62$, $SD = 7.16$) to post-treatment (IPT: $M = 14.62$, $SD = 7.33$, CBT: $M = 12.04$, $SD = 6.98$) across IPT and CBT groups. Whilst reductions in depression scores as measured by the CDI were seen across both groups, significantly greater reductions in scores were reported in those randomised to CBT ($F(1, 107) = 5.96$, $p = 0.016$) compared to IPT. The format of treatment (i.e. group or individual delivery) did not have a significant effect on CDI scores ($F(1, 107) = 1.01$, $p = 0.316$).

A study by Saw and colleagues [47] randomised 20 young people to either culturally-adapted group CBT or a wait-list control. Two depression measures were used—the RADS-2 and the ATQ-Malay—the Malay version of the Automatic Thoughts Questionnaire. There were significant decreases in depression scores in the intervention group from pre-intervention (RADS-2: $M = 80.20$, $SD = 4.83$; ATQ-M: $M = 46.60$, $SD = 3.62$) to post-intervention (RADS-2: $M = 68.90$, $SD = 10.29$; ATQ-M: $M = 33.70$, $SD = 5.54$) with further, more modest, decreases at one-month follow-up (RADS-2: $M = 59.80$, $SD = 14.94$; ATQ-M: $M = 32.50$, $SD = 5.15$)

*Anxiety*. Only one included RCT examined the effectiveness of a culturally adapted intervention for young people with anxiety alone. In the RCT by Khan and colleagues [44], a CBT-based program (Coping Cat) was compared to a wait-list control group with 8–13 year old males (n = 24) who scored >30 on the Screen for Children Anxiety Related Disorders (SCARED). Those receiving the Coping Cat intervention demonstrated significant decreases in scores of anxiety sensitivity ($F = 18.52$, $p < 0.001$), overall anxiety ($F = 11.58$, $p < 0.001$), generalised anxiety ($F = 11.70$, $p < 0.001$), panic disorder ($F = 7.29$, $p < 0.01$) and separation anxiety $F = 6.23$, $p < 0.01$) at 6-months follow-up compared to pre- and post-treatment assessments. Scores on these measures in the wait-list control group remained stable across time points.

*Non-randomised trials (single arm)*. Four of the included studies [48–51, 62] adopted a non-randomised design and included only one treatment arm. All four examined the delivery of a culturally adapted intervention for young people experiencing both anxiety and depression.

In the study by Acaturk et al. [48] all participants received culturally adapted CBT delivered in a group format. Depression was measured using the BDI at pre-treatment, post-treatment and 2-months follow-up. The results demonstrated reductions in depression from 11.9 at pre-treatment to 6.5 at post-treatment and then 5.2 at follow-up. The differences reported across

time points were statistically significant ($F(1.2,14.7) = 8.8$, $p < .01.$). As with depression, anxiety scores, as measured using the SCARED, also reduced across time points from 28.2 at pre-treatment to 19.6 at post-treatment and then to 14.0 at follow-up. Once again, these differences in scores reported across time points were statistically significant ($F(1.4, 16.8) = 19.1$, $p < .001$).

In the study by Goodkind and colleagues [49] all participants received CBT for Trauma in Schools (CBITS). Anxiety was measured using the Multi-Dimensional Anxiety Scale for Children (MASC) which was administered pre-treatment, post-treatment and then at 3 and 6-months follow-up. MASC scores reduced from pre-treatment ($M = 13.29$, $SD = 6.50$) to post-treatment ($M = 9.64$, $SD = 5.07$) and then increased slightly at both 3-month ($M = 10.41$, $SD = 7.12$) and 6-month ($M = 10.34$, $SD = 7.42$) follow-up points. These results demonstrated a significant linear decrease in anxiety scores per 3-month interval of approximately 1 point ($t(75) = 2.15$, $p < .05$). Depression scores, as measured using the CDI, reduced from pre-treatment ($M = 16.06$, $SD = 3.97$) to post-treatment ($M = 14.64$, $SD = 4.52$) and then increased at both 3-month ($M = 15.08$, $SD = 5.21$) and 6-month ($M = 15.44$, $SD = 8.34$) follow-up points. These results suggested a 0.5 change in scores per 3-month interval ($t(22) = 1.98$, $p = .06$).

In the study by Morsette and colleagues [51], all the participants received a manualized CBT intervention delivered in a group format for ten weeks. Three individuals dropped out, leaving four by the end. Depression scores were measured via the CDI and symptoms of PTSD by the CPSS (Child PTSD Symptom Scale)—both at pre- and post-treatment. The average CDI score for the group reduced from pre-treatment ($M = 15$) to post-treatment ($M = 4.5$), with CPSS scores also decreasing ($M = 8.50$ to $M = 3.50$). For one of the young people however, their CPSS score actually increased from pre-to post-treatment and their CDI score remained the same, whereas for another their CPSS score remained the same (although their CDI score significantly reduced).

All participants in the study by Orgilés and colleagues [50] received a Spanish adapted Super Skills for Life (SSL) intervention for the prevention of anxiety and depression. Anxiety was measured using the SCARED at pre-treatment, post-treatment and 12-months follow-up. The results demonstrated that mean SCARED scores decreased across timepoints (pre-treatment: $M = 27.32$, post-treatment: $M = 25.20$, follow-up: $M = 22.55$). As with anxiety, depression was measured at pre-treatment, post-treatment and 12-months follow-up using the CDI. The results demonstrated reductions in scores from pre-treatment ($M = 11.35$) to post-treatment ($M = 9.14$) and then to follow-up ($M = 7.77$). Specifically, whilst 65.3% (n = 77) of participants at baseline had presented with clinically significant symptoms of anxiety and/or depression, this had reduced to 47.3% (n = 53) at post-treatment and then to 40.9% (n = 45) at follow-up.

*Case studies*. The remaining two included studies [52, 53] were case studies of single participants who were experiencing depression. Both studies indicated that the cross-culturally adapted interventions that they delivered showed promising results on alleviating the symptoms of depression. In the study by Binkley and Koslofsky [52] the participant had comorbid Bulimia Nervosa and therefore received Family-Based Therapy for Bulimia Nervosa (FBT-BN) whilst in the study by Duarté -Vélez and colleagues [53] a culturally adapted manual-based CBT was delivered. In the case study by Binkley and Koslofsky, depression was measured using the CDI-2. Whilst the participant's score on the measure was considered to be 'in the very elevated range' at baseline, on sessions two and four the participant's depressive symptoms were in the normal range across all domains on the measure. However, the authors did not report the exact scores for completion of the CDI-2. Similarly, reductions in depression scores were reported in the case study by Duarté -Vélez and colleagues. Here, the participant's depression scores reduced from 58 at baseline on the CDRS-R to 28 at 15-months follow-up

and from 27 at baseline on the CDI to 11 at 15-months follow-up. All results are presented in S3A –S3C Table in S3 File.

## Stakeholder consultation

Alongside the scoping review, we consulted with one adolescent and three parent stakeholders, each of them from an ethnic minority community living in the UK and with lived expertise of depression and/or anxiety (either having experienced this themselves or having looked after someone experiencing this). The purpose of this was to increase our accountability to the members of the communities we are researching, by allowing them the opportunity to dispute or endorse our findings, to add in their own voices and perspectives, and also to help us identify what further research should be prioritised.

Two researchers (PK and MPM) shared the initial findings of the review with the stakeholders and facilitated a discussion. Participants shared their perspectives on the need and acceptability of cross-culturally adapted interventions for the treatment of depression and/ or anxiety in young people and made recommendations for the related interventions The recommendation from the people with lived experience is helpful in informing the design of further research. The discussion was guided by several questions posed by the researchers. Some questions were open-ended, for instance asking participants for their thoughts about the findings from the existing research, whilst others were more focused. The outcomes of this discussion supplemented the review findings and inform the recommendations set out towards the end of this review.

Overall, the participants agreed that cultural adaptation is an important part of an intervention. They concurred with the trend of the conclusions across the included studies that culturally adapted interventions for the treatment of depression and anxiety disorders for adolescents have the potential to be particularly helpful. When discussing the findings with them, the participants felt that the relative lack of engagement with services by this group is due to interventions not being culturally sensitive, for example, they did not include considerations of faith or account for different familial dynamics and values.

The participants identified stigma around mental health and misconceptions about depression/anxiety, as the main barrier in some cultures and communities to seeking support. They stated that culturally tailored elements may have the potential to increase participant engagement; for instance, the inclusion of cultural examples and metaphors could help to bring some of the more abstract and conceptual elements 'alive'. Alongside this, however, further efforts are needed in terms of public, school and community education to improve awareness and understanding and to challenge stigma. Based on the feedback of our stakeholder consultations we found that a 'participatory approach' by engaging different community level stakeholders in the process of adaptation was highly recommended. All of them agreed that it is important to involve parents to increase their awareness of mental health issues in their children and to help optimize the effectiveness of the interventions.

## Discussion

Overall, we identified 17 studies of psychological interventions for young people with depression and/or anxiety disorders that were cross-culturally adapted and considered the appropriate language, metaphors, culturally appropriate terms, and cultural values of the participant. Several cross-cultural adaptation theories and approaches were discussed in the papers. The most commonly used framework, the ecological validity framework [25], was used in six studies. Other cross-cultural adaptation processes were based on the planned adaptation approach, or upon surface and deep structure changes. The differences between these adaptation

frameworks are perhaps more significant on a rhetorical level than on a practical or even a the-
oretical level. Within each framework is a requirement to think both about optimising the
mode of delivery and the nature of the content delivered to the particular culture, setting and
geographic location. Sometimes this means linguistic translation, including literally from one
language into another but also switching to more culturally-resonant metaphors and other
forms of linguistic idiom and symbolism. It can also mean substituting more recognisable and
appropriate examples of people, situations or places that occur in the original content. Some
studies took a non-framework-driven approach, with stakeholder consultations leading to pri-
marily surface-level changes. Most of these studies seemed to adopt a more improvised
approach with iterative consultations between interested parties. Some studies provided little
or no information about the framework or process used for the adaptation.

Overall, the culturally adapted interventions showed promising results. The review found
that apart from one pilot study [42], all studies reported statistically significant improvements
in depression and anxiety symptoms. The failure of the pilot study to result in an improvement
might be in part due to several author-acknowledged limitations to the study including possi-
ble selection bias, a lack of blinding and a small sample size. It may also be that the cultural
adaptation was not applied rigorously enough, as feedback suggested that the younger partici-
pants struggled with the language and metaphors used in the mindfulness intervention.

It is important to note that none of the studies compared a culturally-adapted form of CBT
against the original form of CBT, so it is hard to determine precisely how far a culturally-
adapted approach might have out-performed the original intervention. On the other hand, a
number of studies have shown the lack of effectiveness of non-adapted CBT-based treatments
developed in Western countries in treating those from other cultures with depression and/or
anxiety; for instance a study from 2011 of an un-adapted intervention for persistent depression
amongst Turkish immigrants in Austria [63].

The overall positive picture of cultural adaption drawn from the studies aligns with the
viewpoint of the stakeholder group that we consulted. They highlighted that such adaptations
have the potential to increase the engagement of the participant in intervention delivery and
therefore the intervention's efficacy. It is worth repeating however the group's cautioning that
such an approach needs to go hand in hand with increasing efforts in public, school and com-
munity education to improve awareness and understanding of mental health and to challenge
stigma.

There is little or no consideration in any of the 17 reviewed studies of the cost-effectiveness
of culturally adapted over un-adapted interventions and this is an area of much needed
research. It is likely that, in most cases, adaptation will involve some significant short-term
additional cost, but it is also likely that this would be more than outweighed by the long-term
health and economic benefits.

We found that there is a comparative lack of studies exploring cultural adaptation of an
intervention for a minority community within a (Western) country as opposed to cultural
adaptations of Western-designed interventions to fit another (non-Western) country with a
different majority culture. In our review we identified only two studies in the USA, one pilot
single arm study [49] and one case study [52].

## A participatory approach

Another important observation is that interventions that are developed 'bottom-up', that is
with the involvement of people with lived experience and those delivering the service, tend to
be more effective than 'top-down' approaches and this is consistent with findings in other
meta-analyses [21]. Two of our included studies by Goodkind et al. [49] and Morsette et al.,

2009 [51] used and recommended a community-based participatory approach. The stakeholder group we consulted also advocated for this. Noel et al. [64] recommend Participatory Action Qualitative Research (PAR) methods in adapting an intervention to make it more acceptable to the population of interest. Nicolas et al. [65] emphasise the importance of consultation with different community stakeholders, breaking this down into the following steps: (a) creation of an advisory board, (b) developing a partnership with the community, (c) training the focus group leaders, (d) conducting focus group sessions with adolescents, and (e) integration of focus group data to modify the treatment manual. Using the eight elements of the Ecological Validity and Culturally Sensitive framework [25], the adolescents in their focus group meetings provided informative feedback about the intervention, and evaluated the examples and the language and metaphors used in the intervention manual for accuracy in reflecting the culture of the youth. However, as a comparatively new area of research, further well-designed studies were recommended to show the effectiveness of these processes [65].

## Maintaining treatment fidelity

It can be challenging to culturally adapt a treatment and yet retain fidelity to the essential principles and practices of the intervention. The use of evidence-based therapies (EBTs) and the consideration of the culture, context, and singularity of the person are often framed as antagonistic positions. These seemingly opposing views can however be brought together in the interest of providing optimal care. Duarté -Vélez and colleagues [53] showed that an EBT can be applied with integrity to the core components of the treatment and also with flexibility to the uniqueness of client characteristics. Rosselló and colleagues [40, 41] observed that cultural sensitivity is the ability to entertain both etic (universal norms) and emic (norms for a particular cultural group) and therefore, they proposed that some dimensions must be culturally adapted in a way that is specific to a particular ethnic group of young people where as others are more universal or generic and could therefore apply to other young people as well. This concept of etic and emic norms however only appeared in this study and was only taken so far. As a comparatively new area of research, further work is needed to uncover which elements fall into which category.

## Strengths and limitations

There are some limitations to this review. Due to the heterogeneity of the studies and use of different outcome measures, it was not always possible to synthesize the results well. Owing to the small sample size of most of the included studies, it is difficult to make any definitive claims as to the effectiveness of culturally adapted interventions. As already noted, we did not find any study that allowed us to explore the effectiveness of a culturally adapted intervention versus an intervention which was not culturally adapted in a certain population. We used the main international databases for conducting our search, however, there might be bias in using these databases to report interventions delivered to non-western populations. It was beyond the scope of the study to conduct searches in different regional or country-specific databases and those which included grey literature. Finally, we acknowledge that, in grouping together studies from a wide range of different populations and from different cultures and contexts, there is a risk that we overgeneralise and oversimplify the actual complexity of individual cultural needs.

The review has several strengths. We shed light on a scarce but needed area of research on culturally adapted psychological interventions for young people, which will contribute to the field of evidence-based psychological intervention. We followed a systematic approach in compiling available evidence as well as indicating the effectiveness of culturally adapted

interventions. We also conducted a stakeholder consultation following the scoping review to contextualize the findings and provide validation of recommendations that came from the review.

## Recommendations

We recommend that studies should consider cross-cultural adaptation when looking to apply an intervention to a population in which the intervention was not developed. Studies should identify the dimensions of intervention which are universal or generic and could therefore be applied to any young people and those that should be culturally adapted so they consider or reflect the appropriate language, metaphors, culturally appropriate terms, and cultural values of a specific ethnic group of young people. This will help to make the intervention more acceptable, engaging and effective. The treatment offered for depression and/or anxiety in young people should consider the integration of the best available evidence and consideration of the culture, needs and characteristics of the client. A community-based participatory approach based on a theory-based cultural adaptation framework could be followed in the adaptation process to maximize benefit. We recommend further research on how families, parents, and community representatives and leaders can be involved effectively in the design and development process to optimize the effectiveness of such treatment.

## Conclusion

The review helps to address the lack of existing evidence focusing exclusively on the cross-culturally adapted psychological interventions for the treatment of depression and/or anxiety among young people. We have identified that, on average, culturally adapted interventions to treat depression and anxiety in young people tend to be successful. We have highlighted however that there is a comparative lack of research into interventions for young people with depression and/or anxiety that are tailored to a specific ethnicity group within a country. There is the need for further research to determine, through rigorous empirical testing, which are the most important, clinically effective and cost-effective methods and constituents of cultural adaptation. This is important to help reduce the disparities of mental health on a global scale.

## Supporting information

**S1 File. Search strategy Ovid MEDLINE.**
(DOCX)

**S2 File. Preferred Reporting Items for Systematic reviews and Meta-Analyses extension for Scoping Reviews (PRISMA-ScR) checklist.**
(PDF)

**S3 File.**
(DOCX)

## Author Contributions

**Conceptualization:** Masuma Pervin Mishu, Lina Gega.

**Data curation:** Masuma Pervin Mishu, Philip Kerrigan.

**Investigation:** Masuma Pervin Mishu, Philip Kerrigan, Lina Gega.

**Methodology:** Masuma Pervin Mishu, Lina Gega.

**Visualization:** Masuma Pervin Mishu.

**Writing – original draft:** Masuma Pervin Mishu, Lucy Tindall, Philip Kerrigan.

**Writing – review & editing:** Masuma Pervin Mishu, Lucy Tindall, Philip Kerrigan, Lina Gega.

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
