## [Decision Letter · Decision Letter 0]

14 Jun 2023

PONE-D-22-35714Cross-culturally adapted psychological  interventions for the treatment of depression and/ or anxiety among adolescents: a scoping reviewPLOS ONE

Dear Dr. Mishu,

Thank you for submitting your manuscript to PLOS ONE. After careful consideration, we feel that it has merit but does not fully meet PLOS ONE’s publication criteria as it currently stands. Therefore, we invite you to submit a revised version of the manuscript that addresses the points raised during the review process. Please submit your revised manuscript by Jul 29 2023 11:59PM. If you will need more time than this to complete your revisions, please reply to this message or contact the journal office at plosone@plos.org. Please include the following items when submitting your revised manuscript:A rebuttal letter that responds to each point raised by the academic editor and reviewer(s). You should upload this letter as a separate file labeled 'Response to Reviewers'.A marked-up copy of your manuscript that highlights changes made to the original version. You should upload this as a separate file labeled 'Revised Manuscript with Track Changes'.An unmarked version of your revised paper without tracked changes. You should upload this as a separate file labeled 'Manuscript'.

We look forward to receiving your revised manuscript.

Kind regards,

Muhammad Shahzad Aslam, Ph.D.,M.Phil., Pharm-D

Academic Editor

PLOS ONE

Journal Requirements:

"No funding"

Additional Editor Comments:

Kindly correct the comments given by reviewer and resubmit with rebuttal letter.

Reviewers' comments:

Reviewer's Responses to Questions

**Comments to the Author**

1. Is the manuscript technically sound, and do the data support the conclusions?

Reviewer #1: Yes

Reviewer #2: Yes

Reviewer #3: Partly

2. Has the statistical analysis been performed appropriately and rigorously? 

Reviewer #1: N/A

Reviewer #2: Yes

Reviewer #3: N/A

3. Have the authors made all data underlying the findings in their manuscript fully available?

Reviewer #1: Yes

Reviewer #2: Yes

Reviewer #3: Yes

4. Is the manuscript presented in an intelligible fashion and written in standard English?

Reviewer #1: Yes

Reviewer #2: Yes

Reviewer #3: Yes

5. Review Comments to the Author

Reviewer #1: This is a well written and interesting piece of research which highlights the need for depression and/or anxiety interventions delivered to adolescents to be culturally and contextually sound.

General comments:

While generally well written and intelligible, there are a few grammatical errors in the article which need to be paid attention to. For example, in the Study Participants section, the first sentence does not make sense without a comma after the word ‘studies’. This happens at various occasions, and needs to be corrected, otherwise it is difficult for the reader to understand what the sentence means.

Some abbreviated words are used without their full name in the first use, e.g RCT and USA. USA also needs to be kept for consistency e.g in the discussion, the authors write ‘US’ instead of ‘USA’ which is used earlier in.

Introduction

It would be helpful to define ‘adolescence’ in the introductory section. Also, the authors include studies of those aged 8 to 18 years old. It would be helpful to understand why this age range is included, when adolescence is most commonly defined as 10 to 19 (World Health Organization).

Also, sometimes the word ‘children’ is included, which confuses the reader e.g in the sentence ‘psychological interventions have the potential to improve the mental health of children and adolescents’. I suggest changing the phrasing throughout to young people, as per age criteria and definition, or sticking to adolescents.

Methods

The methods section is very detailed and meets the PRISMA-ScR guidelines. It would be useful to include a PRISMA-ScR checklist in the supplement or figures to show where each item has been included: http://www.prisma-statement.org/documents/PRISMA-ScR-Fillable-Checklist_11Sept2019.pdf

There needs to be some clarity on the recruitment of stakeholders, where they were sampled from, why they were chosen, which culture/country, and ethics used for adolescent involvement. Where the authors wrote ‘participants shared their perspectives on the need and acceptability of cross-culturally adapted interventions and made recommendations’, this needs more clarity. What interventions? Why are their opinions valid or useful here?

For the eligibility criteria, what makes a population or study ‘wrong’? This needs to be better phrased. Also, no restriction is placed on language. How many studies were written in a language other than English, and how did you read these? It would also be good to understand and include a sentence on why grey literature was not included (rationale for eligibility criteria is a key feature of the PRISMA-ScR guidelines). There is bias in using western databases to report interventions delivered to non-western populations, which needs some recognition. The authors could be missing key literature.

For the data extraction section, it would be useful to define ‘any relevant outcome’, as this is quite vague.

Results:

I suggest putting the results/outcomes from Tables 3a and b into Table 1, rather than lots of different tables, as this allows the reader to have all information of each study in one place.

Discussion

The paragraph which says that there is ‘little or no consideration of cost-effectiveness’ of culturally adapted vs. non adapted interventions needs some references. Is there any evidence to support this?

I suggest including a note that the studies were from a range of different populations and cultures/contexts. Your article groups together a wide range of very different populations, which may oversimplify individual cultural needs and be seen as problematic.

Section on ‘A participatory approach’ – do we know why three of the seven young people dropped out of the programme? I think there should be some reference to participatory involvement in your stakeholder interviews and results section, not just in the discussion and conclusion.

Reviewer #2: Thank you so much for choosing me to review this eminent focus review article

The research article gives me holistic knowledge about the psychologically adapted intervention used to treat depression and anxiety among adolescents.

Even though the article's focus is not original, and I read a lot of review papers focusing on the same variables, no one focuses on adolescents, the most crucial developmental stage in human life. See the link below:

https://www.frontiersin.org/articles/10.3389/fpsyt.2020.00212/full

https://www.thelancet.com/journals/lanpsy/article/PIIS2215-0366(23)00118-9/fulltext

https://www.sciencedirect.com/science/article/pii/S0165032720327464

https://pubmed.ncbi.nlm.nih.gov/35168650/

The authors prepared this work very well.

So thank you for your effort; the discussion and conclusion are to the point and reflect what the scoping review papers focused on.

Reviewer #3: The authors summarize current research on cross-culturally adapted psychological interventions for (pre-)adolescents with focus on the treatment of symptoms of anxiety and depression. This is a topic of growing importance, especially bearing in mind that global prevalence rates of mental disorders recently sharply increased in youth populations in connection to Covid-19.

However, there are a number of issues concerning the article‘s main focus of interest , the chosen methodology, and the derived conclusions from the data collected that should be clarified before puplication.

Abtract:

• After revision of the article, an adaptation of the abstract‘s content has to follow

Introduction:

• The authors should pay attention for a consequent use of a cultural sensitive and political correct language. Thus, attributions like „brown“, „black“ and „white“ (p.4) should be appropriately replaced. Same applies to „original country“ (p.5) for which the term „country of origin“ is suggested, instead.

• The theory behind culturally adapting a psychological intervention is just briefly described in the introduction. Instead of introducing common frameworks/models (e.g. Bernal & Sáez-Santiago; „surface vs. deep stractuture adaptations“) within the results, it would be helpful for the reader to give more information about ways of adapting an intervention already within the introduction.

• The authors don’t provide information why they decided for preparing a scoping reveiw instead of a systematic review.

Unfortunately, the study’s main purpose is not clear to the reader, yet. From the abstract and the introduction, it reads as if the authors are primarily interested in examining the effectiveness of culturally adapted interventions for adolescents (although, if this ist he case, the intention could be formulated more concisely). However, going through the discussion, the authors mainly focus and give recommendations on the concrete process and different approaches of culturally adapting an intervention, so that the common thread is missing. A revision in the sense of creating a coherent structure is advised.

Methods:

• The authors provide full information on the algorhythm used for their literature research. For the interest of transparency, it would be helpful, however, if they could explain the underlying principles for their search strategy in a few sentences.

Results:

• Consistent use of italic letters for statistical letters (e.g. „M“, „SD“)

• Table 1: Consistent use of bold letters for headings; consistent, equal structure of table cells (e.g. column „setting and sample“ – please provide information on diagnoses and country of implementation for every study – if unknown, this can be noted accordingly); about adding a column mentioning the kind of cross-cultural adaptation process used by each study should be thought

• Table 3 a-c: These tables all are a bit too confusing for the reader. It should be visible on first sight, which numbers are representing levels of anxiety and which levels of depression. Also, significant changes (pre vs. post as well as treatment- vs. control-group) should be detectable directly for the reader. The column „change reported“(3a)/“significance“ (3b) ist very word-heavy. Table 3c can be omitted completely as it provides almost exclusively text content which can be presented within the article’s written results just as well.

• Figure 2: The benefit from this illustration is arguable as the authors desribe the shown adaptation processes in detail within the results.

• In general, the information provided within the tables 3 a-c are repetitive compared to the written results, to a great extend. Due to the large methodological heterogenity of the reviewed studies, the authors chose a narrative synthesis for the presentation of the studies‘ results. However, an abbreviated and focused display of either the written results or the table contents is advisable.

• p. 23: The authors describe having consulted various stakeholders for discussing their preliminary review findings. From the information shared in the current article, it isn’t clear, however, whether the discussion followed any kind of structured frame or was intentionally kept open. Furthermore, the discussion’s goal and thus the benefit for the current article doesn’t really stand out, so far. Concerning the last paragraph on p. 23, it’s not recognizable whether the authors reproduce the stakeholders‘ points or summarize their own thoughts (which would be misplaced within the results).

Discussion:

• As mentioned above, the discussion’s focus is orientated differently than indicated by abstract and introduction. If the study aims to access the effectiveness of culturally adapted psychological interventions, then the authors have to allow way more space for this topic and discuss their findings critically (e.g. efficacy pre/post, efficacy treatment/non-treatment, (dis-)advantages of different types of control groups, etc.).

• The authors‘ remarks concerning the benefits of a participatory approach when designing culturally adapted interventions was enlightening. However, the initial observation that „bottom-up“ interventions tend to be more effective than „top-down“ interventions cannot be derived from the results that are presented. Similar applies to the conclusion about emic and etic dimensions by Rosselló and colleagues (p.27), which isn’t mentioned in the article before and is accepted as fact by the authors without further discussion or referencing other supporting literature

• Results of the stakeholders discussion (if it shall remain within the article) aren’t discussed at all.

6. PLOS authors have the option to publish the peer review history of their article (what does this mean?). If published, this will include your full peer review and any attached files.

Reviewer #1: **Yes: **Sophia Lobanov-Rostovsky

Reviewer #2: **Yes: **Ayman Mohamed El-Ashry, Lecturer of psychiatric and mental health nursing, faculty of nursing, Alexandria University, Egypt

Reviewer #3: No

---

## [Author Response · Author response to Decision Letter 0]

28 Jul 2023

We would like to thank the peer reviewers for their valuable comments. We are submitting a revised version of the manuscript that addresses the points raised during the review process. The response to all the comments are listed in the attached 'Response to Reviewers' Table.

Response to reviewer 1 comments:

In response to the general comments:

2.We have now added a comma into the ‘study 

participants’ and have identified and 

corrected similar errors.

We have now ensured that all abbreviated 

words are presented in full when first used. 

We have changed ‘US’ to ‘USA’ in the 

discussion for consistency.

Introduction: 3.As suggested by the reviewer, we have

changed the phrasing throughout to ‘young 

people’. However, when describing 

referenced studies, we have used the term 

employed by the study (children and/or 

adolescents).

To clarify this, we have now added the 

following sentence into the abstract: 

‘defined here as children and adolescents 

aged between 8-18 years’ and in the 

introduction.

Methods: 4. We have now added the PRISMA-ScR as 

supplementary file 2. However, the page 

number will need to be revised when the 

article will be published as per the journal’s 

requirement.

5. We have added some additional text to the

‘stakeholder consultation’ section. Ethical 

approval was not required as it was part of 

Patient and Public involvement (PPI), in the 

form of consultation and participants were 

selected from within existing networks.

We clarified, what interventions and why 

their opinions are useful here:

Alongside the scoping review, we consulted 

with one adolescent and three parent 

stakeholders, each of them from an ethnic 

minority community living in the UK and 

with lived expertise of depression and/or 

anxiety (either having experienced this 

themselves or having looked after someone 

experiencing this). The purpose of this was to increase our accountability to the 

members of the communities we are 

researching, by allowing them the 

opportunity to dispute or endorse our 

findings, to add in their own voices and 

perspectives, and also to help us identify 

what further research should be prioritised.

6.We have now added more clarification into 

the section ‘excluded studies’:

‘…,18 were excluded. Reasons for exclusion 

included: having the wrong population (age 

group more than 18 years old) (n=9), wrong 

publication type (conference abstract, 

review, report, editorial, study protocol, any 

grey literature) (n=1), not mentioning 

cultural adaptation/related 

process/effectiveness (n=7), and 

unobtainable full text (n=1).’

One study title was written in Arabic but the 

abstract was written in English. That study 

was excluded at the abstract screening stage 

due to not meeting our inclusion criteria.

We have now acknowledged the limitation 

relating to bias and not including grey 

literature in the discussion section: ‘We 

used the main international databases for 

conducting our search, however, there might 

be bias in using these databases to report

interventions delivered to non-western 

populations. It was beyond the scope of the 

study to conduct searches in different 

regional or country-specific databases and 

those which included grey literature’.

7. We have clarified what outcome data were 

included: 

‘any relevant outcome data (depression 

and/or anxiety)’.

Results: 8. We have moved tables 3a-c (containing detailed 

outcome reports) into a supplementary file

and updated the findings in Table 2. We have 

chosen not to amalgamate the tables as we 

feel this will make the table too busy and 

difficult to follow.

9. Response to comment: 'The paragraph which says that there 

is ‘little or no consideration of cost-effectiveness’ of culturally adapted vs. 

non adapted interventions needs some 

references. Is there any evidence to 

support this?': This statement came from the findings of our 

review as none of the 17 identified studies 

assessed cost-effectiveness, therefore no 

reference was made. We have added some 

text into the discussion:

‘There is little or no consideration in the 

reviewed studies of the cost-effectiveness of 

culturally adapted over un-adapted 

interventions as none of the 17 identified 

studies attempted to assess cost effectiveness and this is an area of much 

needed research’.

10. We have now added this point as one of the limitations of 

the review: ‘Finally, we acknowledge that, in 

grouping together studies from a wide range 

of different populations and from different 

cultures and contexts, there is a risk that we 

overgeneralise and oversimplify the actual 

complexity of individual cultural needs’.

11.

In response to the comment, 'Section on ‘A participatory 

approach’ – do we know why three of 

the seven young people dropped out of 

the programme? I think there should 

be some reference to participatory 

involvement in your stakeholder 

interviews and results section, not just 

in the discussion and conclusion' : Morsette et al. (2009) paper mentioned the 

reasons for why young people dropped out 

and it was not related to the intervention: 

“One participant moved out of the school 

district, one’s parents removed the student 

from the program after child protective 

services were contacted, and one participant

encountered serious medical problems and 

was hospitalized.”

As the reasons for why the young people

dropped out was not related to the 

interventions, we have removed the 

sentence- ‘In spite of this, however, three 

out of seven of the young people dropped 

out of the programme’ from our manuscript.

In the Results section, we added related text.

In the stakeholder’s section we added that: 

‘Based on the feedback of our stakeholder 

consultations we found that a ‘participatory 

approach’ by engaging different community 

level stakeholders in the process of 

adaptation was highly recommended'.

As suggested, we have added participatory 

involvement in the results and stakeholder 

sections, and have revised the related part 

in the discussion.

Reviewer 2:

Many thanks to the reviewer for their encouraging comments. No improvements were

recommended.

Response to the comments of Reviewer 3:

1.We originally used these terms as these were 

used in the original papers that we 

referenced. However, we have now

replaced several words as appropriate.

2.We 

originally used these terms as these were 

used in the original papers that we 

referenced. However, we have now

replaced several words as appropriate.

3. We adopted a scoping review as this is a 

more exploratory approach and the 

research area is still in a preliminary 

stage and very broad. We now added 

related text in the Introduction.

4. In response to the comment, Unfortunately, the study’s main purpose 

is not clear to the reader, yet.: We 

have now clarified the point and made 

the following changes. Objectives:

1. Identify available studies that tested 

the cross-culturally adapted interventions 

for the treatment of depression and/or 

anxiety among young people.

2. to explore the cross-cultural adaptation 

process and frameworks used for the

cultural adaptation.

3. to examine the effectiveness of these 

adapted interventions in the treatment of 

depression and/or anxiety disorders 

among young people.

We have also ensured that these changes 

are reflected in our abstract.

Methods: 5. We have 

added the following section explaining 

the underlying principles for the search 

strategy in the ‘information sources and 

search strategy’ section:

‘The search strategy was developed using 

phrases and keywords of “psychological 

interventions related to depression”, 

“children and adolescent populations”

and “cross-cultural adaptation”. These 

terms were connected using Boolean 

operators and truncations where 

appropriate.’

Results: 6. We have 

made the changes in Table 1. 

We have added information about 

diagnosis for the study by Morsette

(2009).

Country of implementation: Bernal et al. 

(2019), Rossello , Bernal & Rivera-Medina 

(2008) , Morsette (2009), Duarte -Ve lez 

and Bernal & Bonilla (2010).

However, we have not added a column 

mentioning the kind of cross-cultural 

adaptation process used by each study 

because adding more columns will make 

the table too busy and difficult to follow. 

This information has been presented 

separately in Figure 2.

7.We have now moved Tables 3 a-c to a 

supplementary file. Owing to this, we felt 

it important to retain Table 3c for 

consistency. Therefore, we present the 

following: Table 3(a) for reporting RCTs, 

3(b) for reporting non-RCTs and 3(c) for 

reporting case studies. 

Where possible we have condensed the 

information presented in the tables and 

have presented all anxiety measures in 

italics to make it easier for the reader to 

identify the different measures.

8.As this 

Figure provides an idea of cultural 

processes used in the included studies at 

a glance, we felt it important to retain this 

Figure for readers who prefer a visual 

presentation of information. 

9. As previously mentioned, we have now 

presented Tables 3 (a-c) within a 

supplementary file. This can be viewed 

by readers if they prefer to see results 

presented in a tabulated format. We have 

however condensed the information 

presented within these tables.

10. We have 

now added some additional sentences to 

the text:

‘Some questions were open-ended, for 

instance asking participants for their 

thoughts about the findings from the 

existing research, whilst others were 

more focused’.

Discussion: 11.As per our response to comment number 

4, our updated objectives now 

correspond to what is presented in the 

discussion.

We have 

now added the following section relating 

to the participatory approach:

‘However, as a comparatively new area of 

research, further well-designed studies 

are needed to show the effectiveness of 

these processes’.

We have also updated the e following 

sentences about emic and etic 

dimensions:

‘This concept of etic and emic norms 

however only appeared in this study and 

was only taken so far. As a comparatively 

new area of research, further work is 

needed to uncover which elements fall 

into which category’.

We have now added sentences based on

stakeholder consultation findings 

presented in the discussion section:

‘The overall positive picture of cultural 

adaption drawn from the studies aligns 

with the viewpoint of the stakeholder 

group that we consulted. They 

highlighted that such adaptations have 

the potential to increase the engagement 

of the participant in intervention delivery

and therefore the intervention’s efficacy. 

It is worth repeating however the group’s 

cautioning that such an approach needs 

to go hand in hand with increasing efforts 

in public, school and community education to improve awareness and 

understanding of mental health and to 

challenge stigma.’

Response to academic editor’s comment:

1.The manuscript meets PLOS ONE's style requirements and file naming.

2.We addressed the financial disclosure by stating that: “The authors received no 

specific funding for this work.”

3. We added captions for our Supporting Information files at the end of your manuscript, 

and update any in-text citations to match accordingly

---

## [Decision Letter · Decision Letter 1]

13 Aug 2023

Cross-culturally adapted psychological interventions for the treatment of depression and/or anxiety amongyoung people: a scoping review

PONE-D-22-35714R1

Dear,

We’re pleased to inform you that your manuscript has been judged scientifically suitable for publication and will be formally accepted for publication once it meets all outstanding technical requirements.

Kind regards,

Muhammad Shahzad Aslam, Ph.D.,M.Phil., Pharm-D

Academic Editor

PLOS ONE

Additional Editor Comments (optional):

Reviewers' comments:

Reviewer's Responses to Questions

**Comments to the Author**

1. If the authors have adequately addressed your comments raised in a previous round of review and you feel that this manuscript is now acceptable for publication, you may indicate that here to bypass the “Comments to the Author” section, enter your conflict of interest statement in the “Confidential to Editor” section, and submit your "Accept" recommendation.

Reviewer #1: All comments have been addressed

Reviewer #3: All comments have been addressed

2. Is the manuscript technically sound, and do the data support the conclusions?

Reviewer #1: Yes

Reviewer #3: Yes

3. Has the statistical analysis been performed appropriately and rigorously? 

Reviewer #1: Yes

Reviewer #3: N/A

4. Have the authors made all data underlying the findings in their manuscript fully available?

Reviewer #1: Yes

Reviewer #3: Yes

5. Is the manuscript presented in an intelligible fashion and written in standard English?

Reviewer #1: Yes

Reviewer #3: Yes

6. Review Comments to the Author

Reviewer #1: No concerns. All my comments have been addressed in this revision. It is an important and interesting piece of work, which is methodologically sound.

Reviewer #3: Dear authors, thank you for your thorough revision of the manuscript. Well noted, that all issues have been adressed. This version of the manuscript is acceptable for publication.

7. PLOS authors have the option to publish the peer review history of their article (what does this mean?). If published, this will include your full peer review and any attached files.

Reviewer #1: No

Reviewer #3: No

---

## [Editor Report · Acceptance letter]

22 Aug 2023

PONE-D-22-35714R1 

Cross-culturally adapted psychological interventions for the treatment of depression and/or anxiety among young people: a scoping review 

Dear Dr. Mishu:

I'm pleased to inform you that your manuscript has been deemed suitable for publication in PLOS ONE. Congratulations! Your manuscript is now with our production department. 

Kind regards, 

on behalf of

Dr. Muhammad Shahzad Aslam 

Academic Editor

PLOS ONE